

# Past ice sheet-seabed interactions in the northeastern Weddell Sea Embayment, Antarctica

Jan Erik Arndt[1,2], Robert D. Larter[2], Claus-Dieter Hillenbrand[2], Simon H. Sørli[3], Matthias Forwick[3], James A. Smith[2], Lukas Wacker[4]

[1]Alfred Wegener Institute Helmholtz Centre for Polar and Marine Research, Am Handelshafen 12, 27570 Bremerhaven, Germany
[2]British Antarctic Survey, High Cross, Madingley Road, Cambridge CB3 0ET, United Kingdom
[3]Department of Geosciences, UiT The Arctic University of Norway, Postboks 6050 Langnes, N-9037 Tromsø, Norway
[4]ETH Zürich, Laboratory of Ion Beam Physics, Schafmattstrasse 20, CH-8093 Zurich, Switzerland

*Correspondence to*: Jan Erik Arndt (Jan.Erik.Arndt@awi.de)

**Abstract.** The Antarctic Ice Sheet extent in the Weddell Sea Embayment (WSE) during the Last Glacial Maximum (LGM; ca. 19-25 calibrated kiloyears before present, cal. ka BP) and its subsequent retreat from the shelf are poorly constrained, with two conflicting scenarios being discussed. Today, the modern Brunt Ice Shelf, the last remaining ice shelf in the northeastern WSE, is only pinned at a single location and recent crevasse development may lead to its rapid disintegration in the near future. We investigated the seafloor morphology on the northeastern WSE shelf and discuss its implications, in combination with marine geological records, for reconstructions of the past behaviour of this sector of the East Antarctic Ice Sheet (EAIS), including ice-seafloor interactions. Our data show that an ice stream flowed through Stancomb-Wills Trough and acted as the main conduit for EAIS drainage during the LGM. Post-LGM ice-stream retreat occurred stepwise, with at least three documented grounding line still stands, and the trough had become free of grounded ice by ~10.5 cal. ka BP. In contrast, slow-flowing ice once covered the shelf in Brunt Basin and extended westwards toward McDonald Bank. During a later time period, only floating ice was present within Brunt Basin, but large 'ice slabs' enclosed within the ice shelf occasionally ran aground at the eastern side of McDonald Bank, forming ten unusual ramp-shaped seabed features. These ramps are the result of temporary ice-shelf grounding events buttressing the ice further upstream. To the west of this area, Halley Trough very likely was free of grounded ice during the LGM, representing a potential refuge for benthic shelf fauna at this time.

## 1 Introduction

In Antarctica, the largest uncertainty for the grounding line position of the ice sheet at the Last Glacial Maximum (LGM) exists in the Weddell Sea Embayment (WSE; The RAISED Consortium et al., 2014). Two conflicting scenarios were proposed for the reconstructed position of the LGM grounding line: In one scenario, which is predominantly based on marine geological and geophysical information, the grounding line is located near the shelf edge (Hillenbrand et al., 2012, 2014; Larter et al., 2012). The alternative scenario is mainly based on cosmogenic nuclide exposure dates on terrestrial samples from the eastern and southwestern hinterland of the WSE, which, in combination with ice sheet modelling, indicate only limited thickening of





the East Antarctic Ice Sheet (EAIS) during the LGM, and reconstructs a grounding-line position about 650 km further south in Filchner Trough (Hillenbrand et al., 2014; Bentley et al., 2010; Hein et al., 2011). However, newly published terrestrial data from around the WSE indicate thickening of the EAIS during the LGM, and that previous terrestrial studies were influenced

by cosmogenic nuclide inheritance due to surface preservation caused by coverage with cold-based ice, lending support to the first scenario (Nichols et al., 2019). Additional marine geophysical and geological data led to the hypothesis that grounding line fluctuations were very dynamic in Filchner Trough and that a (re-)advance to 75°30'S on the outer shelf occurred during the Early Holocene (Arndt et al., 2017). This relatively late grounding line advance was explained by ice-flow switching in the hinterland that resulted in redirected ice flow from the West Antarctic Ice Sheet (WAIS) into Filchner Trough, which

previously was covered by grounded ice draining the EAIS. A possible trigger for this flow switch is a temporarily different EAIS/WAIS development, with an earlier EAIS retreat allowing WAIS drainage into Filchner Trough and grounded ice reaching its maximum extent thereafter. Hence, understanding the Filchner Trough paleo-ice dynamics necessitates improved understanding of the EAIS retreat history, i.e. from the poorly studied WSE.

Marine geological and geophysical data offshore from the Luitpold Coast, just east of Filchner Trough (Fig. 1 inset), suggest

post-LGM grounding line retreat of individual glaciers and ice streams draining the EAIS there progressed from north to south between ~12.8 cal ka BP and ~8.4 cal ka BP (Hodgson et al., 2018). The radiocarbon age constraints, however, were generally poor and, for some sediment cores, not very reliable because of the apparent presence of reworked biogenic material (organic carbon and calcareous microfossils) in the dated sediment samples. Subsequently, the grounded glaciers and their ice shelves persisted as long as their fronts were buttressed by the advanced Filchner paleo-Ice Stream and its floating ice shelf or their

ice-shelf bases were pinned on bathymetric highs adjacent to the over-deepened glacial troughs (Hodgson et al., 2018).

Further to the northeast, the Brunt Ice Shelf and the Stancomb-Wills Glacier Tongue form the nearest extant large ice-shelf system fed by the EAIS east of Filchner Trough (Fig. 1). Radiocarbon dates from core IWSOE70 3-7-1 recovered just in front of Brunt Ice Shelf (Figs. 2) were interpreted to indicate that this location was not overrun by grounded ice since at least 32.5 cal ka BP (Stolldorf et al., 2012; see Table S2 for correction and calibration of original reported AMS [14]C ages). However, an

alternative scenario in agreement with the available age constraints suggests that grounded ice overran the core site sometime between 32.5 cal ka BP and 20.7 cal ka BP (Hillenbrand et al., 2014). Sometime after the LGM, the Brunt Ice Shelf is assumed to have been grounded on multiple "smaller-scale topographic highs" of the McDonald Bank (Fig. 1) that acted as pinning points stabilizing the ice shelf (Hodgson et al., 2019).

Today, the Brunt Ice Shelf is undergoing significant changes. Recent crevasse and crack formation and their growth may lead

to widespread loss of bed contact with the only remaining pinning point, McDonald Ice Rumples (Fig. 1), which may result in rapid ice shelf disintegration in the near future (Hodgson et al., 2019; De Rydt et al., 2018). The permanent British research station Halley (Fig. 1) that had operated continuously since 1956, was relocated in 2017 and subsequently has remained unoccupied during the last three Austral winters due to the increased threat of ice-shelf disintegration. Glaciological surveys revealed the unusual structure of the Brunt/Stancomb-Wills ice shelf system, which in fact consists of three sectors: Brunt Ice

Shelf in the west, Stancomb-Wills Glacier Tongue in the east, and a sector in between these two ice shelves that we here refer



to as the 'suture zone' (Thomas, 1973; Hulbe et al., 2005). The Brunt Ice Shelf is composed of blocks of meteoric ice of variable sizes (i.e., a number of individual icebergs) that were "cemented" together by sea ice and snow accumulation immediately after their calving (King et al., 2018). In the suture zone, larger tabular icebergs consisting of meteoric ice, up to 25 km across, that we refer to as 'ice slabs' are interspersed with and bound together by thick perennial sea-ice, while the

Stancomb-Wills Glacier Tongue consists mainly of meteoric ice (Thomas, 1973; Hulbe et al., 2005). Fluctuations in flow velocity of the Brunt Ice Shelf were observed over the last 50 years, most likely resulting from variations in the mechanical connection to the McDonald Ice Rumples pinning point (Gudmundsson et al., 2016). Knowledge of the past development of this ice shelf system and its interaction with pinning points is crucial for a better understanding of its complex nature, which is a prerequisite for modelling and forecasting its future development.

In this study, we investigate the seabed geomorphology of the northeastern WSE shelf offshore from the Brunt Ice Shelf system (Fig. 1) with the purpose of elucidating the imprints of past ice sheet development in this region. Newly mapped submarine glacial landforms in combination with data from recently collected and previously published marine sediment cores provide valuable clues for past ice dynamics and reveal both the ice drainage pattern during the LGM and that slow-flowing, cold-based ice covered a large area. We interpret a set of large seafloor ramps as products of the temporary grounding of meteoric-

ice slabs that were locked within perennial sea ice and pushed forward by ice shelf motion. During these temporary grounding events, the influence of the growing ramps on ice drainage progressively increased as they acted as pinning points, with the grounded ice slabs buttressing ice flow further upstream.

## 2 Methods

### 2.1 Swath bathymetry

Swath bathymetric data were acquired in the study area during 16 Antarctic expeditions with RV *Polarstern* and RRS *James Clark Ross* between 1985 and 2018 using various acquisition systems (see Table 1). All bathymetric data were sound velocity corrected using conductivity-temperature-depth (CTD) measurements and post-processed to remove outliers in CARIS Hips & Sips or in MB-System (Caress et al., 2019; Caress and Chayes, 1996), respectively. All data were exported to ASCII XYZ format and, subsequently, jointly gridded at 25 m resolution in QPS Fledermaus with a weighted moving average algorithm.

Bathymetric data were visualised and analysed in QPS Fledermaus and ESRI ArcGIS.

### 2.2 Acoustic sub-bottom profiling

Sub-bottom profiler data were acquired during RV *Polarstern* expeditions PS82, PS96, and PS111 and during RRS *James Clark Ross* expedition JR244. On RV *Polarstern*, a Teledyne RESON Parasound System DS3 (P70) was used to acquire the sub-bottom profiles. This system uses the parametric effect of two high primary frequencies to generate a resulting secondary

low frequency, which is used to profile sub-seafloor features. The secondary low frequency usually was set to ~4 kHz. The



resulting beam width of the signal was 4.5°. On RRS *James Clark Ross*, a Kongsberg Topographic Parametric Sonar (TOPAS) PS 018 system was used, which works in a similar way. Two primary frequencies around 18 kHz were used to generate a 'chirp' secondary transmission pulse with frequencies between 1.3 and 5 kHz. The beam width of the system is 5°. The penetration depth of both systems depends on the local seafloor characteristics, e.g. when the seabed consists of soft, fine-

grained sediments the signal can penetrate down to about 200 m below the seafloor surface with a depth resolution of approximately 30 cm.

## 2.3 Marine sediment cores

Four sediment cores (GC634, GC635, GC636 and GC637) recovered during RRS *James Clark Ross* expedition JR244 in March 2011 using a 3 m long gravity corer with a diameter of 110 mm were analysed for this study (Supplementary Table S1).

In the following sections, we exclusively focus on results from core GC635 (latitude 74° 59.5'S, longitude 25° 27.8' W, water depth 494 m, recovery 1.16 m) because it was the only core that provided an age constraint for the time of bedform formation. Magnetic susceptibility, P-wave velocity and wet-bulk density were analysed on whole and split core sections using GEOTEK multi-sensor core loggers (MSCLs) at the British Ocean Sediment Core Research Facility (BOSCORF; Southampton, UK) and at the Department of Geosciences, UiT The Arctic University of Norway (Tromsø, Norway), respectively. After splitting

the cores, their working and archive halves were described visually and using smear slides at the British Antarctic Survey (BAS; Cambridge, UK). Sedimentary structures were recorded visually and using X-radiographs, which had been obtained at 2 cm intervals from the archive halves with a GEOTEK MSCL-X-ray computed tomograph at UiT (MSCL-XCT; voltage of ~120 kV, current of ~225μA). Sediment colour was recorded on the archive halves visually using Munsell Soil Colour Charts (Munsell Color Company, Inc., 2010) at BAS and using a Jai L-1070CC 3 CCD RGB Line Scan Camera with a resolution of

70 μm mounted to an Avaatech XRF core scanner at UiT. Shear strength measurements were conducted every 5-10 cm on the working halves of the cores using a hand-held shear vane. 1 cm thick discrete half-round samples at selected depths (intervals from 5-20 cm) were taken from the working halves of the cores to determine (i) the water content by weighing the samples before and after freeze-drying, and (ii) the grain size distribution (i.e., contents of gravel, sand and mud) by wet- and dry-sieving over 63 μm and 2 mm. The sand fractions were subsequently investigated under a microscope for analysing the

composition of the coarse fraction and picking calcareous microfossils. The picked microfossils were AMS [14]C dated using the Mini Radiocarbon Dating System (MICADAS; Synal et al., 2007; Wacker et al., 2010) at the Laboratory of Ion Beam Physics of the Eidgenössische Technische Hochschule (ETH; Zürich, Switzerland). The resulting dates were corrected for a regional marine reservoir effect of 1215±30 years established by the uncorrected [14]C-age of bryozoans in seafloor surface sediments at site PS1418-1 (see Hillenbrand et al., 2012) and calibrated using the CALIB 7.1 software (Reimer et al., 2013;



Stuiver and Reimer, 1993). Previously published AMS [14]C-dates on benthic foraminifera from piston core IWSOE70 3-7-1 (Anderson & Andrews, 1999; Stolldorf et al., 2012) were corrected and calibrated accordingly (Supplementary Table S2).

### 2.4 Ice-slab tracking

Satellite imagery and a map (Figure 2 of Thomas, 1973) were used to trace the movement of four large, meteoric ice slabs enclosed within the suture zone from 1973 to 2017. The map of Thomas (1973) provides the earliest record of their locations.

This map was manually georeferenced in ESRI ArcGIS using geographic coordinates from the graticule of the map as reference points. Satellite imagery is from Landsat missions, courtesy of U.S. Geological Survey. Three representative dates were chosen that document the ice slab movements: 1986-01-30 (Landsat 5), 2000-01-06 (Landsat 7) and 2017-02-04 (Landsat 8).

## 3 Results

### 3.1 Seafloor morphology

An overview of the high-resolution bathymetry data in the study area is shown in Figure 1. Three major bathymetric features on the shelf are distinguished: Two bathymetric depressions located on the innermost shelf that we refer to as Brunt Basin (in the South) and Stancomb-Wills Trough (in the North), and a T-shaped ridge with a SSW-NNE oriented long axis that we refer to as McDonald Bank (Fig. 1). The McDonald Bank separates both depressions from Halley Trough to the west. The ESE-ward pointing branch of its 'T' crossbar partly separates the depressions from each other. This branch also includes the

shallowest point on the bank at 196 m water depth. The shallowest mapped part of the bank further south is ~210 m water depth at a location close to the Brunt Ice Shelf front and just about 10 km west of the McDonald Ice Rumples. This water depth matches sub-ice shelf measurements further east which indicate that the seafloor depth at the McDonald Ice Rumples is about 212 m (Hodgson et al., 2019). The top of the bank elsewhere lies at an average water depth of 260 m. The NW-directed branch of the crossbar at the seaward end of McDonald Bank separates Stancomb-Wills Trough from Halley Trough and forms its

deepest part (>300 m water depth, Fig. 1).

The maximum surveyed water depth in Brunt Basin is ~690 m at a location that is presently covered by the re-advanced Brunt Ice Shelf. In contrast to Brunt Basin, Stancomb-Wills Trough extends to the continental shelf edge. The trough increases in depth inshore from ~500 m water depth at the continental shelf edge to ~700 m water depth at the most landward surveyed shelf location, approximately 50 km from the shelf edge. Single-beam echosounder data, acquired at a time when the ice-shelf

front had a more easterly position, show that deepening continues further inland (Arndt et al., 2013). The slope along the trough axis is ~0.2°.

Numerous smaller-scale submarine bedforms are present in the study area. We categorize these into nine classes (A-I) based on their morphology. The classes and their morphologic properties are described in detail in Table 2 and their spatial distribution is shown in Figure 2. Figures 3–7 show their morphologies in detail.



The most prominent bedforms in the study area are ten ramp-shaped features of Class H (Figs. 3, 4). Their shapes vary, but generally their steep flanks face towards the west. Ramps 1-4, located on the eastern side of McDonald Bank, are crescent-shaped and have a width of about 8-10 km at their open ends and crest widths of about 3 km (Fig. 2 ,3). In contrast,  ramps 5-10, lack a clear crescent shape (Fig. 4). In cross-profiles, ramps 1-10 appear to have a shape similar to that of the wedges of Class G (Fig. 5b, c, and d), but the ramps appear in a different bathymetric setting, outside of a cross-shelf trough. In addition,

their front and back slope angles generally differ from those of the Class G wedges, which have a mean front slope of about 2.5° and a mean back slope of only about 0.5°. In contrast, mean frontal and back slopes of ramps 1-4 are 3-5° and 1.5-2.2°, respectively (Fig. 5b), and those of ramps 6 and 8-10 are ~5-8° and 0.7-1°, respectively. Two exceptions are ramps 5 and 7 that have slopes (front slope ~2° and back slope ~0.1-0.5°) similar to those of the Class G wedges. However, both ramps have the same orientations and occur in the same bathymetric setting as the other Class H ramps. Some ramps are super-imposed

on other ramps, e.g. ramp 9 formed on top of ramps 8 and 10 (Fig. 4), and the northern part of ramp 2 overlies the southern part of ramp 1 (Fig. 3). Many of the ramps have irregular morphologies at their tops, where the seafloor surface undulates with an amplitude of less than 10 m, e.g. ramps 1 and 2 (Fig. 3).

## 3.2 Core lithology and chronology

Most coring operations carried out in the study area resulted in no or very little recovery, and only six sediment cores with a

recovery of more than 20 cm are available (Fig. 2 and Supplementary Table S1). Coring operations were unsuccessful especially on McDonald Bank, i.e. on and close to Class H ramp 2, where four coring attempts failed. Seafloor photographic imagery of Class H ramp 10, acquired by the Ocean Floor Observation System (OFOS) during expedition PS96 at station 10-3 (Piepenburg, 2016), shows that the seafloor is predominantly covered by fine-grained mud (Supplementary Fig. S1). Where this fine-grained sediment is absent, a gravel layer is observed, suggesting that the mud forms only a thin and locally absent

veneer on top of the gravel layer. Elsewhere, cobbles and boulders are locally present on the seafloor. Gravel layers are difficult to penetrate with conventional coring devices, and together with the observed cobbles and boulders, seriously hamper or even prevent sediment recovery by coring. Under the assumption that the OFOS imagery shows the typical seafloor of McDonald Bank and Class H ramps, this may explain the limited core recovery there.

In Stancomb-Wills Trough, core recoveries were higher than in Brunt Basin and on McDonald Bank (Fig. 2). However, except

for neighbouring cores GC634 and GC635 (Fig. 2), calcareous material suitable for obtaining reliable AMS $^{14}$C ages was absent (e.g. Hillenbrand et al., 2013). Core GC634 was 14.5 cm long and provided a paired Late Holocene age of ~4.9 cal ka BP obtained from calcareous benthic and planktic foraminifera (Supplementary Table S2). Of all cores with recoveries higher than 20 cm, exclusively the cores GC635 (this study) and IWSOE70 3-7-1 (Anderson & Andrews, 1999; Stolldorf et al., 2012) provided radiocarbon age constraints useful for reconstructing the glacial history of the study area (Supplementary Table S2).

Core GC635 was retrieved from the southern edge of Stancomb-Wills Trough at a location where Class E bedforms are present (Fig. 2 and 4). The sedimentary sequence consists of a thin layer of homogenous sponge bearing, gravelly, sandy mud near the core top (0-9 cm), overlying a poorly sorted muddy diamicton (9-116 cm) (Fig. 8a). The sand content  varies slightly throughout



the core. The diamicton is laminated and stratified between 9 and 70 cm core depth, and massive below that. A large pebble is present at 60-65 cm depth influencing the physical properties in this core interval. Shear strength increases and water content decreases down-core within the upper 70 cm of the diamicton, although there is a local minimum in shear strength at 68.5 cm core depth. Increased shear strengths and low water contents characterise the diamicton below 70 cm depth, documenting higher compaction than in the upper part of the diamicton. One bivalve shell fragment (2.7 mg) at 50 cm depth in core GC635 was AMS [14]C-dated. The sample (laboratory code: ETH-69709) provided an uncorrected age of 10,475±90 [14]C years and after marine reservoir effect correction and [14]C-age calibration an age of 10,520±260 cal. years BP (Supplementary Table S2).

## 4.    Morphological interpretation

### 4.1.    Iceberg ploughmarks (Class A)

We interpret the single, linear to curvilinear furrows with random orientations predominantly occurring on the outer shelf part of Stancomb-Wills Trough as iceberg ploughmarks. Iceberg ploughmarks are common seafloor features on the Antarctic continental shelf (e.g. Gales et al., 2016; Lien et al., 1989). Grounding of iceberg keels rework the upper few meters of the seabed, forming a berm of iceberg-turbated sediments at the ploughmark edge (Fig. 6c). Pre-existing seafloor bedforms may have been eradicated in areas with abundant ploughmarks.

### 4.2.    Iceberg ploughmarks with preferred orientation (Class B)

Class B bedforms have similar morphological attributes as the iceberg ploughmarks of Class A (see Tab. 2 and Fig. 6). However, in contrast to the latter, they have a preferred, although not perfectly parallel, orientation. We interpret these features as ploughmarks carved into the seafloor by numerous single and probably some multi-keeled icebergs. The direction of these ploughmarks is aligned sub-parallel to the modern and past ice-sheet/-shelf flow directions, and their occurrence is restricted to an area within Stancomb-Wills Trough landward of the continental shelf edge, with the water depths of the aligned ploughmarks (500 – 600 m) being greater than that of the shelf edge (~500 m and less). Icebergs eroding these features could not have drifted into the area from elsewhere and must, thus, have originated from an ice-shelf present in Stancomb-Wills Trough. We assume that calved icebergs were pushed by the ice shelf front in its flow direction, probably enclosed within a newly formed sea ice melange that held them in place. If the drafts of the iceberg keels were sufficiently deep, this resulted in ploughmarks with preferred orientation as observed in our data. A similar process has previously been suggested for parallel multi-keeled iceberg ploughmarks in the outer shelf sections of Filchner Trough in the WSE (Larter et al., 2012) and Cosgrove-Abbot Trough in the easternmost Amundsen Sea Embayment, West Antarctica (Klages et al., 2015).

### 4.3.    Mega scale glacial lineations (Class C and D)

The parallel, linear ridges of Classes C and D located in the deeper parts of the Stancomb-Wills Trough resemble mega-scale glacial lineations (MSGLs). MSGLs are common features in formerly glaciated areas and form along the direction of ice flow



at the base of fast-flowing ice streams (Clark, 1993; King et al., 2009) within a deformable sediment layer (Alley et al., 1986; Ó Cofaigh et al., 2005; Reinardy et al., 2011). The orientation of Class C and D MSGLs differs by 15-25° (Figs. 2, 6), and the

amplitudes of Class D MSGLs are smaller than the amplitudes of Class C MSGLs. This suggests that the two classes of MSGLs were produced at different times, with the more subdued Class D MSGLs located further offshore being older than the Class C MSGLs. This is supported by acoustic sub-bottom profiler data: Class D lineations are characterized by a thicker or more diffuse top reflector than the more distinct top reflector of Class C lineations, which could indicate a thin sedimentary drape on Class D lineations that was deposited when the Class C lineations were being formed (Fig. 6c).

**4.4.     Squeezed and smeared ridges of grounded tabular icebergs enclosed in perennial sea-ice (Class E)**

The elongated Class E ridges are present only within the most landwardshelf area at the base and locally on top of the ice-shelf proximal wedge-shaped ramps of Class H (Fig. 2 and 4). The amplitude and shape of the ridges strongly resembles that of MSGLs, especially locally where some of them occur as sets of parallel ridges. However, unlike the MSGLs of Classes C and D, all the ridges of Class E are curvilinear rather than linear, or even have a more irregular, sinuous shape and their directions

in general are more variable. Occasionally, the ridges also terminate in a crescent-shaped 'dam' (Fig. 4), similar to terminal berms of icebergs ploughmarks (e.g. Lewis et al., 2016). Sub-bottom profiler data across the ridges reveal their formation in an acoustically transparent sediment layer on top of a strong, continuous sub-bottom reflector (insets of Fig. 4). Incisions into the sub-bottom reflector are absent. This contradicts an interpretation as a typical iceberg ploughmark berm that usually is formed by material excavated from the ploughmark centre (for example see Classes A and B in Fig. 6c). However, the smeared

impression of the features may suggest a similar kind of formation.

Considering all morphologic properties of the ridges, these are neither typical glacial lineations nor typical iceberg ploughmarks. Their proximity to wedge shaped ramps of Class H and their orientation in dip direction of these ramps might indicate that the formation process of these bedforms is related. We hypothesize that these ramps were formed by temporary grounding of large meteoric ice slabs enclosed within perennial sea ice in the suture zone between Brunt Ice Shelf and

Stancomb-Wills Glacier Tongue (see below). In agreement with this hypothesis, we suggest that the ridges of Class E were formed by seafloor sediments being squeezed into basal crevasses in the ice slabs, when these ran aground. Subsequently, these sediments were 'smeared' into the direction of the movement of the ice slabs. A source for these sediments may be the till layer, in which the MSGLs (Class C and D) in Stancomb-Wills Trough had formed (Fig. 6c). This would also explain why these ridges are only present close to the trough, where such soft till substrate is available (Fig. 4), but that they are absent on

the ramps further to the south in Brunt Basin (Fig. 3), i.e. at a location distal from the MSGLs.

**4.5.     Lateral marginal-moraine (Class F)**

The wedge shaped bank on the outer shelf that forms the south-western flank at the seaward end of Stancomb-Wills Trough is a lateral moraine. Batchelor and Dowdeswell (2016) reviewed 70 lateral moraines described in previous studies and identified two different types that can be distinguished by their geometry: lateral shear-moraines and lateral marginal-moraines. Lateral





shear-moraines are either roughly symmetrical or have a steeper trough-facing flank in cross-section. In contrast, lateral marginal-moraines are wedge-shaped in cross profile with a steeper flank facing away from the trough. They are effectively grounding zone wedges formed by spreading of the downstream part of an ice stream that was not laterally constrained. We observe the latter in Stancomb-Wills Trough (Fig. 5a) indicating that the Class F feature formed as a lateral marginal-moraine.

### 4.6.  Grounding-zone wedges (Class G)

The three wedge-shaped mounds of Class G located in the inner part of Stancomb-Wills Trough (Figs. 4, 6a, 6c) are interpreted as grounding-zone wedges (GZWs). GZWs are ice-marginal deposits formed at the transition from a grounded ice stream to a floating ice shelf, the so-called grounding zone (e.g., Alley et al., 1989; Batchelor and Dowdeswell, 2015). These bedforms are produced by high subglacial sediment supply under fast flowing ice streams to their termini during phases of grounding line still stands (Ó Cofaigh et al., 2008). A GZW typically has a steep front and a gentle back slope due to the restricted

accommodation space beneath the ice shelf close to the grounding line. The mapped MSGLs (Classes C and D, Figs. 2, 6) give evidence for past ice streaming in Stancomb-Wills Trough that facilitated high subglacial sediment transport. The northernmost of the three identified GZWs is located in the central part of Stancomb-Wills Trough, with sub-bottom profiler data indicating that the GZW was deposited on top of a prominent reflector, most likely representing a pre-existing seafloor surface (left inset in Fig. 6c). The other two GZWs, located further upstream, are only mapped at the southern side of the trough, but likely

continue into the trough centre that is presently ice shelf covered. The observed set of GZWs indicates that at least three phases of grounding line still stands occurred within the trough during the retreat of the paleo-ice stream.

### 4.7.  Iceberg ramps (Class H)

On first examination, the wedge-shaped ramps of Class H show similarities to GZWs, but their mean front and back slopes are steeper, with the exceptions of ramps 5 and 7 that also have similar crest heights as the northernmost GZW in Stancomb-Wills

Trough (see Fig. 5b, c, d). In addition, the location of ramp 10 at the southern edge of Stancomb-Wills Trough raises the possibility that this ramp was deposited as a lateral marginal-moraine (Class F) by the paleo-ice stream when it flowed within the trough. However, the close proximity, orientation, and morphological similarity to the other ramps points to formation of ramps 5, 7 and 10 by the same process as the other Class H ramps.

Indications for the presence of a paleo-ice stream south of Stancomb-Wills Trough at the location of the ramps are lacking.

Only such an ice stream would have been able to supply large volumes of subglacial sediment to the grounding zone fast enough to build up a GZW (Batchelor and Dowdeswell, 2015). Glacial lineations, which would indicate fast ice flow (e.g. King et al., 2009), are absent, both on the tops and in front of Class H ramps (Figs. 3, 4). In addition, the ramps are located outside a cross-shelf trough, the typical topographical setting of GZWs where repeated channelized ice streaming over several glacial stages deeply eroded the seabed, providing a plentiful source of sediment (e.g. Livingstone et al., 2012). Consequently,

we exclude the possibility that the ramps were formed as GZWs.



The most prominent ramps are the crescent-shaped ramps 1-4 at the eastern edge of McDonald Bank (Figs. 2, 3). In contrast to ramps 5-10, which exhibit multiple examples of superimposition, of the four remaining ramps only ramps 1 and 2 overlap marginally. Hence, ramps 1-4 probably are more suitable for understanding the process responsible for their formation.

Arcuate terminal moraines located at fjord mouths in the Arctic have heights comparable to the ramps on the northeastern WSE shelf and have a similar crescent shape, e.g. in Raudfjorden, northern Svalbard (Ottesen and Dowdeswell, 2009), or in Scoresby Sund, East Greenland (e.g. Arndt, 2018). The crescent shape of these terminal moraines is a result of diverging ice flow due to disappearing lateral topographic constraints at the fjord mouth. In contrast, the discussed ramps in our study area are located in areas without suchtopographic constraints. Therefore, we conclude that, even though some morphological similarities exist, the ramps are features other than terminal moraines.

Hill-hole pairs observed on the Norwegian continental shelf have similar arcuate shapes and dimensions as ramps 1-4 (Rise et al., 2016). Hill-hole pairs are supposed to occur in areas of slow flowing ice, which locally froze to the bed, ripped-up seafloor sediment and subsequently redeposited the sediment "lump" further downstream and glacitectonics deforming the deposit. A similar glaciological regime of slow ice flow probably existed in the area of the Class H ramps, which is characterized by the absence of glacial lineations and a location outside a cross-shelf trough. The holes of the hill-hole pairs on the Norwegian shelf

mark the locations from where the sediment hills were excavated, and are characterised by clearly distinguishable depressions of rugged terrain that are about 15 m deeper than the surrounding seafloor. Such delineated holes or depressions are absent upstream from the ramps identified in our study (Figs. 3, 4). However, bathymetric profiles across ramps 1-3 show that their back-slopes have a 'listric' topography, becoming relatively flat beyond about 5 km upstream of their fronts (Fig. 5b). This may indicate that some kind of excavation process was active at this location, as the slope gets steeper towards the crest of the

ramp. However, the seafloor surface in these probably excavated areas is smooth rather than rugged, whilst a distinct boundary typical for the depressions of the hill-hole pairs on the Norwegian shelf is missing. Furthermore, hill-hole pairs with a dimension of more than five kilometres seem to be rare, with the ones of the Norwegian shelf being the only published examples. In contrast, hill-hole pairs documented in the Antarctic are smaller (Klages et al., 2013, 2015; Larter et al., 2019). Therefore, we conclude that the process of cold-based hill-hole pair formation is unlikely to explain the formation of the Class

H ramps.

Bedforms on the western flank of Crary Bank in the Ross Sea were described as 'wedges pinned on seamounts' and were interpreted to represent grounding line retreat positions of a re-grounded ice shelf (Fig. 5b in Greenwood et al., 2018). These features share a similar geographical setting as the Class H ramps as both are located on the slope of a shallow bank that faces the grounded ice sheet and have similar heights. Greenwood et al. (2018) observed subtle lineations on top of the wedges in

the Ross Sea. Such lineations are absent from the tops of the Class H ramps on the northeastern WSE shelf. The only lineations observed here are the curvilinear ridges of Class E, which exclusively occur at the base of the easternmost ramps. The Crary Bank wedges are mostly ≤4 km across, with only one elongated feature extending over a length of approximately 7 km, so they are generally smaller than that of the Class H ramps. Unlike the ramps, the features in the Ross Sea are furthermore circular rather than crescent shaped. In addition, their tops are flat over a distance of about 2 km. In contrast, bathymetric





profiles across Class H ramps show that their tops are convex (Fig. 5b,c). The Crary Bank mounts are located in the volcanically active region of the Terror Rift and were interpreted as subglacial volcanic features (Lawver et al., 2012). The northeastern WSE shelf, however, lacks any Cainozoic volcanic activity. The morphological differences and the absence of recent volcanic activity thus suggest a different formation process for the Class H ramps.

The smoothness of the seafloor at and around most of the ramps suggests that it consists of sedimentary material. In contrast,
the rugged terrain described inland of ramp 6 may represent a bedrock substrate, which geographically coincides with the inferred location of a large mafic intrusion (Jordan and Becker, 2018). When drifting icebergs run aground on a sedimentary seafloor substrate with sufficient force, they produce ploughmarks with berms on either side (see also Class A bedforms). If an iceberg encounters seafloor substrate where drag associated with ploughing increases to match the force pushing the iceberg into the seabed, the movement of the iceberg slows down drastically and eventually stops, with the iceberg keel producing a
terminal berm that consists of the pushed substrate at its front (e.g. Lewis et al., 2016; Jakobsson et al., 2011). A study from the eastern Amundsen Sea Embayment shelf shows that most of the observed iceberg ploughmarks there have widths of 40-300 m and incision depths of less than 20 m (Wise et al., 2017), with iceberg ploughmark berm heights typically being about 50% of the incision depths. Hence, their dimensions are significantly smaller than those of the Class H ramps (more than a magnitude narrower and less than 20% tall), suggesting that the ramps are morphological features other than typical iceberg
ploughmarks.

Nevertheless, the crescent shape of ramps 1-4 resembles the shape of a typical terminal iceberg berm well, but at a considerably larger scale (Fig. 3). The structure of the Brunt/Stancomb-Wills ice-shelf system and reconstructed ice flow orientation changes indicate that larger icebergs enclosed in perennial sea-ice melange driven by greater force likely grounded in the area in the past. Therefore, we assume that these enclosed icebergs once grounded with sufficient force to form the Class H ramps by
deformation of materials on the seafloor. At the same time, these ramps would have become increasingly active as pinning points buttressing ice flow further upstream. We outline the glaciological circumstances and the hypothesized formation mechanism of the Class H ramps in more detail in the discussion.

### 4.8.    Ice-marginal ridges and grooves (Class I)

The slightly curvilinear ridges and grooves of Class I occur only on top and at the western flank of McDonald Bank (Fig. 7).
Their orientation is at a high angle or even perpendicular to flow directions indicated by the long axes of the nearby paleo-ice stream troughs, i.e. Halley Trough to the west and Stancomb-Wills Trough to the north. Orientations of subglacial bedforms in the latter suggest an inter ice-stream environment. For such an environment, most of the morphological attributes of the Class I ridges (amplitude, width, length, orientation) indicate that these features are retreat moraines as observed in other inter-ice stream areas of previously glaciated continental shelves (Ottesen and Dowdeswell, 2009).

Sub-bottom profiles across the ridges and grooves show predominantly a continuous single seafloor-surface reflector, locally incised by very small (approx. 1 m and less) furrows (Fig. 7b). Sub-bottom reflectors are absent. In contrast, the deeper seafloor of Halley Trough directly NW of the ridges and grooves shows a different acoustic facies (left zoom-in of Fig. 7b). Here,





acoustically transparent lenses overlie a continuous, flat sub-bottom reflector. As MSGLs are apparently absent in Halley Trough (this study; Gales et al., 2014), we do not consider that these acoustically transparent lenses correspond to a soft till layer like that observed in other Antarctic paleo-ice stream troughs (e.g. Ó Cofaigh et al., 2005; Livingstone et al., 2012). In contrast, several iceberg ploughmarks are mapped in this area (Fig. 2). Accordingly, we interpret the transparent lenses in Halley Trough as deposits resulting from iceberg turbation that were formed by ploughing of iceberg keels through relatively soft sedimentary strata. The change in acoustic facies towards the ridges and grooves in the southeast indicates that different environmental conditions prevailed, i.e. with grounded ice covering the seafloor at and to the southeast of the ridges and grooves but being absent from Halley Trough.

It remains, however, unclear why the Class I features consist of ridges, typical for retreat moraines, and grooves. Nevertheless, based on the reasons outlined above, we suggest that these ridges and grooves were formed close to the terminus of slow-flowing ice at some time in the past. Remarkably, some of the grooves are incised within ramps 1 and 3, implying that they must have formed after these Class H ramps.

## 5.    Discussion

### 5.1.    Iceberg ramp formation

The comparison of Class H features to other similar bedforms described in the literature does not allow the identification of an unequivocal formation process for these bedforms (see Section 4.7). Here, we propose a ramp formation mechanism that is strongly linked to the unusual structure of the Brunt/Stancomb-Wills ice-shelf system and past ice flow orientation changes, resulting in moulding of these ramps into McDonald Bank by large ice slabs enclosed within the suture zone of the ice shelf system.

The Brunt Ice Shelf has an unusual ice shelf structure because, unlike other ice shelves, the inland ice here loses its structural integrity when flowing across the grounding line and breaks apart into icebergs, which subsequently are 'glued' together by freezing sea ice and drifting snow to form the ice shelf (King et al., 2018). A similar process is also active in the suture zone located between the slower flowing Brunt Ice Shelf and the faster flowing Stancomb-Wills Glacier Tongue (Fig. 1, Thomas, 1973; Gudmundsson et al., 2016). The different velocities result in shearing and the production of large ice slabs just downstream of the grounding line. Satellite imagery shows that these icebergs, enclosed within the thick sea ice/snow drift today, have lateral dimensions from 6 to 35 km (Fig. 9a), which is within the size range of the observed ramps. Today, the icebergs trapped within the multiyear sea ice are up to about 300 m thick as shown by the Bedmap2 dataset (Fretwell et al., 2013) and a single transect of Operation IceBridge (www.nsidc.org/data/icebridge). Accordingly, the modern draft of these icebergs is about 275 m. Iceberg keels probably reached greater water depths during glacial stages when sea level was lower and ice sheets probably were thicker. Hence, we assume that the water depths of 200 to 440 m, at which the ramps occur today (Fig. 5b and c), are within the range of past iceberg-keel drafts.



Today, the transport direction of the enclosed icebergs is approximately towards northwest, providing an iceberg trajectory
that passes just about 10 km north of the northernmost ramps (Fig. 9a, b). Under glacial-time conditions, however, different
ice flow patterns may have redirected the trajectory of the enclosed icebergs towards the ramps, most likely as a result of
increased ice flow through Stancomb-Wills Trough. The MSGLs mapped in Stancomb-Wills Trough (feature classes C and
D) indicate that such increased outflow occurred in the past and that the ice flow direction was different. The ice flow direction
change is, furthermore, supported by the subglacial topography further upstream. A large NNE-directed subglacial trough is
located upstream of the grounding line of Stancomb-Wills Glacier Tongue (Fig. 9b, Fretwell et al., 2013). This trough is on
average 40 km wide and 700 m deep, and extends at least 200 km upstream of the grounding line, with today's ice flow in its
centre exceeding velocities of 300 m/yr close to the grounding line (Mouginot et al., 2017). Ice flowing through this trough
today feeds into the Stancomb-Wills Glacier Tongue that flows approximately in a NW-ward direction (Fig. 9b). Upstream of
the suture zone and upstream of Brunt Ice Shelf similar subglacial troughs and ice flow velocities are lacking (Fig. 9c).This
indicates that the subglacial trough upstream of the Stancomb-Wills Glacier Tongue is the main gateway for ice discharge at
present, and probably acted as such during times of an extended EAIS, too. The MSGLs mapped in Stancomb-Wills Trough
were formed during these times and their orientation is approximately WNW-ESE, i.e. they point upstream in the direction of
where the subglacial trough reaches the modern grounding line. Despite a major data gap underneath the ice shelf, the latest
sub-ice shelf bathymetry model calculated from gravity inversion (Hodgson et al., 2019) indicates that this subglacial trough
probably is a continuation of Stancomb-Wills Trough. These findings suggest that past ice flow in the trough area was
redirected counter-clockwise in a WNW direction and, thus, forced the trajectory of large icebergs enclosed in the suture zone
also into a WNW direction, i.e. exactly into the direction of the ramps on the eastern flank of McDonald Bank (Fig. 9b).

The morphological properties of the ramps support a formation process by iceberg grounding and subsequent moulding. Apart
from the fact that their shape resembles that of terminal berms of iceberg ploughmarks, but at a larger scale, at least some of
the ramps show indications of super-imposing by neighbouring ramps. One example is the northern berm of ramp 2, which
super-imposes the southern berm of ramp 1 (Fig. 3). Another example is ramp 9, whose flank overlies the edges of ramps 8
and 10 (Fig. 4). In addition, this ramp shows a curvilinear long axis that is aligned to the direction of modern flow of the Brunt
Ice Shelf (De Rydt et al., 2018). Super-imposed iceberg berms are observed frequently on heavily scoured shelves of glaciated
continental margins (e.g. Bjarnadóttir et al., 2016). The areas of irregular seafloor morphology on top of some ramps may be
explained by the iceberg formation scenario as well. The amplitude of such irregularities is <10 m, which is well within the
range of usual iceberg ploughing. The irregular surface was probably formed at the time when the large iceberg broke apart,
as this break-up would have resulted in locally variable pressure exerted onto the ramps by free floating, grounded, and
capsizing iceberg fragments.

Even though until now neither seafloor bedforms of similar dimensions, shapes and in similar bathymetric settings have been
observed, nor a recent equivalent of the formation process has been reported, the sum of the observations detailed above
supports the hypothesis that the ramps were formed by icebergs enclosed in a perennial sea-ice melange. In Figure 10, we
illustrate the proposed formation process of a hypothetical, roughly 3 km long ramp in an about 6 year-long time series based



on the modern-day ice shelf flow velocity of approximately 500 m/a. The illustration also provides a concept of stressesthat
we infer were active during ramp formation, and which would have affected the integrity of an ice slab and surrounding

perennial sea ice. These stresses comprise back stress caused by friction induced by ice-slab grounding and gravitational stress
caused by the progressive 'jacking up' of the ice slab front. At a certain stage, these stresses may reach thresholds leading to
a stress release by fragmentation of the sea-ice melange or fracturing of the ice slab due to the cliff height at its front edge
exceeding its yield strength.

### 5.2. Glacial reconstruction of the northeast Weddell Sea Embayment

### 5.2.1. Chronological constraints

The absolute chronology of ice sheet history in the study area is poorly defined by only two existing chronological constraints
from marine sediment cores (Fig. 8, Table S2): One is from core GC635, which was recovered from the southern edge of
Stancomb-Wills Trough, and the other is from core 3-7-1, which was collected from the eastern flank of McDonald Bank near
the present Brunt Ice Shelf front (Fig. 2; Stolldorf et al., 2012; Anderson et al., 1980; Anderson et al., 1981).

We assign the sediments recovered in core GC635 to three different lithological facies (Fig. 8a): The gravelly sandy mud
between 0 and 9 cm core depth bears abundant sponge spicules indicating deposition of this facies under (seasonal) open
marine conditions. The lamination and stratification of the predominantly terrigenous diamicton from 9 to 70 cm suggests that
this facies was also deposited in a glacimarine setting, which is supported by the presence of a bivalve shell fragment. The
predominantly terrigenous composition of this facies together with the absence of bioturbation point towards  deposition

underneath an extended ice shelf or under perennial sea-ice cover. In contrast, the purely terrigenous, homogenous diamicton
from 70 to 116 cm core depth is characterised by high shear strength and wet-bulk density and low water content. This facies
is consistent with subglacial deposition as a soft till. According to our morphological interpretation, site GC635 is located in
the area of Class E ridges, which were formed by large grounded tabular icebergs enclosed in perennial sea ice at some time
after initial retreat of grounded ice, and lies on the back-slope of the innermost GZW in Stancomb-Wills Trough (Figs. 2, 4).

Therefore, we assume that the soft till retrieved in the basal part of core GC635 formed as part of the grounded tabular iceberg
ridges (Class E). A bivalve shell fragment found in the glacimarine diamicton overlying the soft till provided an AMS [14]C date
of 10.5 cal. ka BP. Accordingly, this age can be considered as both a minimum age for retreat of grounded ice from site GC635
and a chronological constraint for the time when the site was still covered by an ice shelf.

Core 3-7-1, recovered near the McDonald Ice Rumples, provided five AMS [14]C dates obtained from calcareous benthic

foraminifera and one AMS [14]C date from an echinoid spine for ice-proximal glacimarine diamictons overlying subglacial till.
The resulting ages are >52.8 [14]C ka BP (echinoid spine) and between 32.5 and 14.6 cal ka BP (benthic foraminifera) (Fig. 8b;
Stolldorf et al., 2012; Anderson et al., 1980; Anderson et al., 1981). However, the radiocarbon dates showed down-core age
reversals that document sediment reworking (Fig. 8b). Therefore, these ages can either be interpreted that grounded ice had
retreated from site 3-7-1 at some time before ca. 32.5 cal. ka BP (Stolldorf et al., 2012), or that grounded ice overran this site





between ca. 32.5 cal. ka BP and 20.7 cal. ka BP (see Fig. 5 in Hillenbrand et al., 2014) and that the subglacial till deposited during this advance was subsequently reworked by iceberg scouring.

To improve the understanding of the depositional environment at site 3-7-1, we tried to relate the core location to its surrounding seafloor morphology. Core 3-7-1 was recovered during the last of three International Weddell Sea Oceanographic Expeditions in 1970 (IWSOE70). At this time, ships' positioning systems were not as capable as today due to the absence of satellite-based positioning systems, or position fixes being several hours apart and having much larger uncertainties than modern GPS systems. Thus, the coordinates provided for the core location may be erroneous. In contrast, the precision of water depth measurements with single-beam echo sounders was already quite high in the 1970s. This is especially the case in shallow waters as depth accuracy is usually a function of water depth. Therefore, we examined, if the water depth given for core site 3-7-1 matches the water depth of our bathymetric data near this site. The water depth of 235 m measured for site 3-7-1 on expedition IWSOE70 is 25 m shallower than the water depth of 260 m inferred for the core location from our bathymetric data. The closest seafloor with shallower water depth is located further to the south and, therefore, we assume that the 'true' core site is located there, about 3 km south of the originally reported core location (Fig. 2). We also analysed the position of another core, core 013 of expedition IWSOE68, which was recovered in the area just two years earlier in 1968 and was described as containing abundant benthic foraminifera (Anderson et al., 1981). The measured water depth at site 013 was 507 m, but the water depth from our swath bathymetry data is only 230 m at this location. The closest known area with water depths deeper than 500 m is in Brunt Basin, i.e. at a distance of at least 12 km. This highlights that core locations determined in the pre-satellite navigation era must be treated with caution. Nevertheless, the assumed 'true' sampling site of core 3-7-1 is located in open water conditions today, but just about 4 km offshore from the McDonald Ice Rumples, thus, in close proximity to modern grounded ice. Therefore, we think it is very likely that this core site was covered by grounded ice at some time during the LGM. This grounded ice may have occurred due to either a thicker ice shelf and/or lower sea level, or by an advance of the EAIS grounding line.

### 5.2.2. Glacial history of Stancomb-Wills Trough

During the time of an extended EAIS, most ice in the study area was discharged via ice streaming through Stancomb-Wills Trough, as documented by the mapped MSGLs, rather than through Brunt Basin. The ice-flow direction inferred from these lineations indicates that the ice stream originated from a known subglacial trough, which continues 200 km upstream of the modern grounding line, and which we assume is the upstream continuation of Stancomb-Wills Trough (Fig. 8a). Such trough systems usually have been deepened by subglacial erosion beneath ice streams during multiple glacial cycles. Therefore, the subglacial lineations indicate that Stancomb-Wills Trough acted as a conduit for a major paleo-ice stream draining the EAIS. The lateral marginal-moraine west of the trough on the outer shelf proves that the Stancomb-Wills paleo-ice stream once reached the shelf edge. Lateral marginal-moraines are suggested to be formed in a similar way as GZWs, but in areas of ice stream flow divergence, i.e. areas without lateral constraints established by either subglacial topography or neighbouring zones of slow-flowing grounded ice (Batchelor and Dowdeswell, 2016). Thus, the area southwest of the lateral marginal-moraine



must have been free of grounded ice, allowing ice stream divergence and probably the development of an ice shelf that extended from the steep distal side of the lateral moraine. This scenario is also in accordance with our interpretation of the curvilinear

ridges and channels on the western side of McDonald Bank (Class I features) to represent bedforms created at the margin of slow ice-sheet flow. The north-eastern extension of this slowly flowing ice area coincides with the south-eastern extension of the lateral marginal-moraine (Fig. 2), indicating that further inshore the Stancomb-Wills paleo-ice stream was laterally constrained by slow ice flow at its south-western edge in Brunt Basin.

The GZWs located in Stancomb-Wills Trough provide evidence of at least three still stands during grounding line retreat.

According to the minimum deglaciation age from core GC635, located in the area of the grounded tabular iceberg ridges (Class E) on the back-slope of the innermost GZW (Fig. 4), these phases occurred at some time before 10.5 cal ka BP. The orientation of the very subtle glacial lineations (Class D bedforms) offshore from the northwesternmost GZW differs by 15° to 25° when compared to the orientation of the other glacial lineations (Class C bedforms) further inshore. This indicates that the ice flow direction changed after this grounding line still stand.

After grounding line retreat landward of the southeasternmost GZW by 10.5 cal ka BP, icebergs with drafts reaching up to a modern-day water depth of 600 m calved from the front of the Stancomb-Wills paleo-ice stream as is documented by the maximum water depth of iceberg ploughmarks in the trough. Furthermore, the nearly parallel orientation of the innermost ploughmarks aligned with the trough's long axis shows that the iceberg trajectories were to some extent constrained. Probably, the icebergs were enclosed within sea ice, with the ice shelf motion pushing the icebergs further offshore.

The ice shelf itself was most probably pinned for some time on the easternmost and shallowest part of the ESE-pointing branch of the  T-crossbar of McDonald Bank (196 m minimum water depth) that at this time acted as an ice shelf pinning point similar as the McDonald Ice Rumples today. Hence, this shoal is even shallower than the McDonald Ice Rumples, which are less than 220 m deep (Hodgson et al., 2019). Therefore, an extended ice shelf, even with modern-day thickness, would be able to ground on the shoal and, thus, be buttressed by the ice grounded on this pinning point.

### 5.2.3.    Glacial history of Brunt Basin and McDonald Bank

In contrast to Stancomb-Wills Trough, landforms indicative of fast flowing ice, i.e. MSGLs, are lacking in Brunt Basin and on McDonald Bank. Hence, at least during the most recent glaciation, no paleo-ice stream flowed through Brunt Basin. Instead, the formation of the Class H ramps by large tabular icebergs and of the Class I ridges and grooves by ice-marginal processes suggests that two other glaciological regimes affected this area during the past. The first regime was characterised by a phase

of slowly flowing, grounded ice covering Brunt Basin up to the western flank of McDonald Bank. Retreat of this slowly flowing, grounded ice from McDonald Bank likely was rapid as further ice marginal landforms indicative of step-wise retreat, i.e. moraines, are absent on the eastern flank of the bank. However, processes active in the second regime may have reworked the seabed sediments, including possibly recessional moraines, subsequent to grounded ice retreat. This second glaciological regime was characterised by presence of floating ice within Brunt Basin. Thick, large ice slabs enclosed in perennial sea ice



within an ice shelf, similar to the modern day situation in the suture-zone between Brunt Ice Shelf and Stancomb-Wills Glacier Tongue, moulded ramps into the eastern flank of McDonald Bank.

There are barely any chronological constraints for these two regimes, including their relative temporal order (see section 5.2.1). Some grooves of Class I seem to incise into the western parts of ramps 1 and 3 (Fig. 3). This suggests that at least the formation of these two outer ramps pre-dates the phase of slow grounded ice flow on McDonald Bank. However, according to our interpretation that individual thick icebergs formed the ramps, these events themselves likely occurred intermittently over an extended period of time. This may indicate that the formation of some ramps occurred before and the formation of other ramps after the phase of slow grounded ice flow in Brunt Basin. Either way, the ramps show that the Stancomb-Wills/Brunt ice shelf system was repeatedly buttressed and, therefore, to some degree 'stabilized' by McDonald Bank in a similar way as the modern Brunt Ice Shelf is now buttressed by the McDonald Ice Rumples. As a consequence of decreasing ice drainage through Stancomb-Wills Trough, the transport direction of icebergs enclosed in the suture zone shifted northwards and ice shelf extent may have decreased. At some point, these icebergs did not reach McDonald Bank anymore, leaving the modern McDonald Ice Rumples as the only pinning point for the ice shelf until today.

The absence of a lateral topographic constraint for the Stancomb-Wills paleo-ice stream on the outer shelf as suggested by the lateral marginal-moraine (Class F) is in line with the observation of slow, grounded ice flow on McDonald Bank as far north as the limit of the ridges and grooves (Class I) observed on its western flank. Thus, the geographic relation of these two bedform classes suggests that their formation coincided. This also implies that the area offshore from the ridges and grooves, i.e. Halley Trough, was free of grounded ice during this phase of glaciation. Therefore, Halley Trough may represent a refuge, where benthic shelf communities on the Weddell Sea shelf survived in-situ the LGM (Barnes and Hillenbrand, 2010).

## 6.    Conclusions

The glacial morphology on the northeastern WSE shelf shows that in times of an extended EAIS most ice was discharged through Stancomb-Wills Trough, which probably extends about 200 km upstream of the modern-day grounding line. Retreat of grounded ice in the trough was step-wise, with at least three phases of grounding line still stands, during which GZWs were formed in the trough. Thereafter, an ice shelf was present in the trough, as documented by the almost parallel orientation of linear iceberg ploughmarks in the deep outer section of the trough.

In Brunt Basin and on McDonald Bank, the existence of two other glaciological regimes is inferred from the data. One regime was characterised by slowly-flowing, grounded ice that covered Brunt Basin westward up to a set of ridges and grooves on the western edge of McDonald Bank. At the same time, grounded ice was absent in Halley Trough, suggesting that this trough may have been a refuge for benthic shelf fauna during the LGM. The other regime was characterised by the presence of floating ice within Brunt Basin. Large 'ice slabs' enclosed and steered by perennial sea ice, similar to the modern day situation within the suture zone between Brunt Ice Shelf and Stancomb-Wills Glacier Tongue, ran aground on the eastern side of McDonald Bank and were subsequently pushed seaward by ice-shelf advance to mould ramps into the seafloor. Such ramps have not been

observed elsewhere on formerly glaciated margins, probably because their formation requires a unique ice-shelf setting as it is provided by the Brunt/Stancomb-Wills ice-shelf system. The ramps imply that on several occasions in the past the Stancomb-Wills/Brunt ice shelf system was repeatedly buttressed and 'stabilized' by icebergs pinned on McDonald Bank. After a northward shift of the iceberg trajectories, the modern McDonald Ice Rumples remained the only pinning point of the ice shelf until today.

The chronological constraints of ice sheet retreat in the study area remain sparse. Nevertheless, our data provide a new minimum age for grounded ice retreat from inner Stancomb-Wills Trough by 10.5 cal ka BP. For Brunt Basin, the question of whether the area was overrun by grounded ice during the LGM remains unresolved. Core 3-7-1 is open to alternative interpretations regarding this question due to down-core age reversals and, as shown here, the unclear geographical core location and, thus, an unclear geomorphological context.

*Data availability.* The compiled AWI/BAS bathymetric data are available via PANGAEA: 10.1594/PANGAEA.907173. Sub-bottom profiler data of expeditions PS82, PS96 and PS111 can be requested from AWI via PANGAEA: 10.1594/PANGAEA.837893, 10.1594/PANGAEA.860442, and 10.1594/PANGAEA.897301. TOPAS sub-bottom profiler data are available from the UK Polar Data Centre on request. Full results of analyses on sediment cores GC634 to GC637 are available from C.-D. H. on request.

*Author contributions.* J.E.A., R.D.L., and C.-D.H. developed the concept and led the writing of this paper. J.E.A. was responsible for compilation of used bathymetric data sets. C.-D.H., S.H.S., M.F., J.A.S, and R.D.L. were responsible for the recovery and investigation of the marine sediment cores, with S.H.S. carrying out most of the laboratory analyses. L.W. carried out the MICADAS radiocarbon dating. J.E.A, R.D.L., C.-D.H., S.H.S., M.F. and J.A.S. interpreted the geomorphological and core data. All authors contributed to the writing of the manuscript.

*Competing interests.* The authors declare no competing interests.

*Acknowledgements.* We thank all captains, crews and scientists supporting and enabling acquisition of the scientific data and samples used in this study. We thank Hilmar Gudmundsson for sharing pre-selected Landsat imagery of the study area with no or only little cloud cover. J.A. was funded by the Deutsche Forschungsgemeinschaft (DFG, German Research Foundation) grant AR 1087/1-1. This study is part of the Alfred Wegener Institute Helmholtz Centre for Polar and Marine Research program Polar Regions and Coasts in the Changing Earth System (PACES II), the BAS program 'Polar Science for Planet Earth'. S.H.S. and M.F. thank Aker BP ASA for financial support.





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





**Figure 1: Overview map of the study area showing color-coded high-resolution bathymetric data and geographic locations mentioned in the text. Greyscale contour lines are in 100 m interval from the IBCSO V1.0 data set (Arndt et al., 2013). Background satellite image is from Landsat, courtesy of the U.S. Geological Survey. Locations of other figures and two bathymetric cross profiles, shown in Fig. 5, are indicated by dashed rectangles and lines, respectively. The inset map shows the study area in a broader Antarctic context (background from Arndt et al., 2013). CC = Caird Coast; EAIS = East Antarctic Ice Sheet; FRIS = Filchner Ronne Ice Shelf; FT = Filchner Trough; LC = Luitpold Coast; WAIS = West Antarctic Ice Sheet.**







**Figure 2: Mapped landform classes in the study area, for location see Fig. 1. Dots and crosses mark the locations of piston cores (PC) and gravity cores (GC) (for core site details, see Supplementary Table S1). Note that cores 13 and 3-7-1 are presumably mislocated due to inaccurate positioning capabilities in the 1970s, when these cores were retrieved. Core site 3-7-1 is relocated by shifting it to the nearest location in the high-resolution bathymetry data that has the same water depth as measured at the core location. The red square indicates the area where seafloor images were taken by the Ocean Floor Observation System (OFOS; see Fig. S1).**



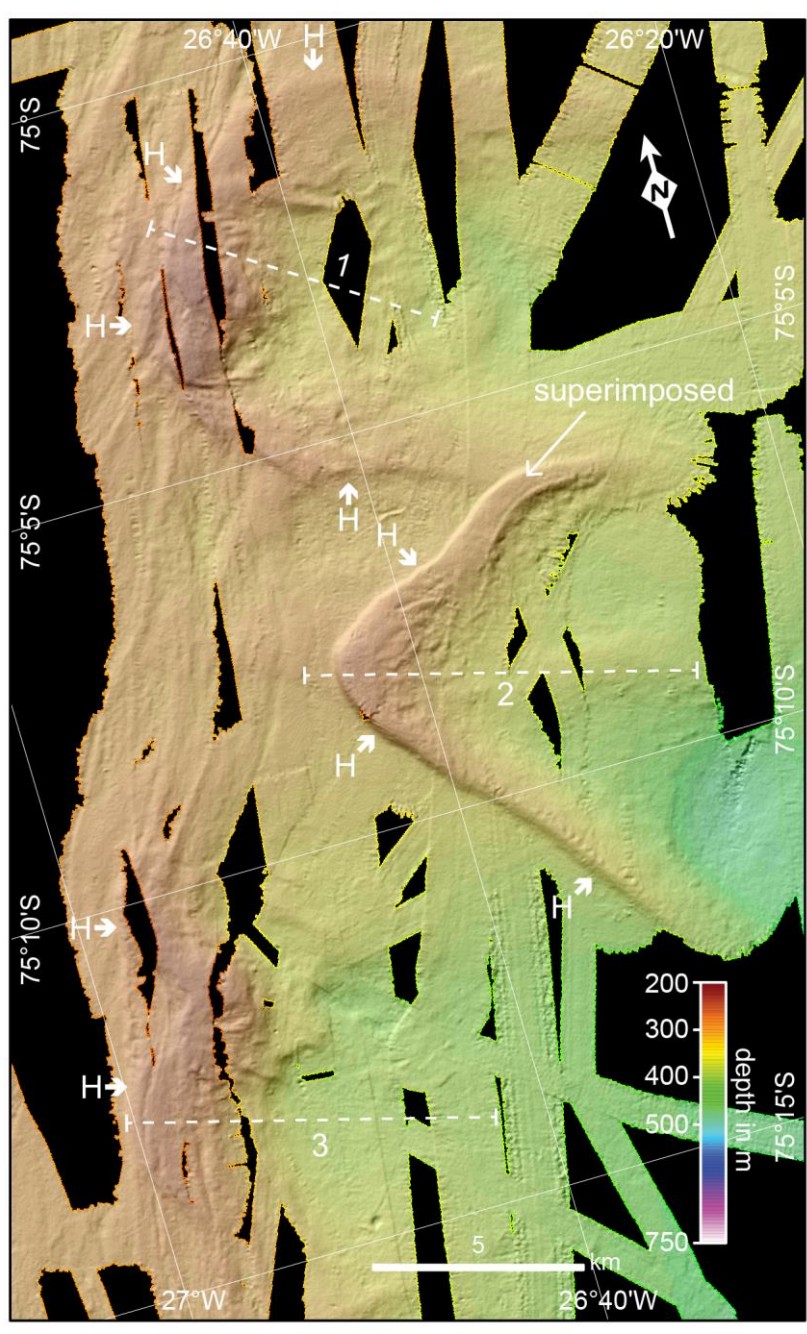

**Figure 3: High-resolution bathymetric map of the McDonald Bank showing crescent-shaped ramps of Class H (fronts marked by bold short arrows), note that the northern part of ramp 2 superimposes on the southern part of ramp 1 and the locally irregular seafloor morphology on the top of ramp 2. Dashed numbered lines indicate locations of ramp profiles shown in Fig. 5.**







**Figure 4: High-resolution bathymetric map from the easternmost part of McDonald Bank showing parallel ridges squeezed by and smeared at the base of grounding ice slabs enclosed in perennial sea ice (long dashed arrows, Class E) and wedge shaped ramps (fronts marked by bold short arrows, Class H). Note the more rugged terrain inshore of the ramps that may reflect outcrops of a mafic intrusion. Bold dashed line marks the front of the innermost grounding-zone wedge (Class G). Thin dashed numbered lines indicate locations of ramp profiles shown in Fig. 5. Inset figures shows acoustic subbottom profiles across Class E beforms.**







**Figure 5: Bathymetric profiles across (a) the lateral moraine of morphological Class F, (b) Class H crescent -shaped ramps 1-4, (c) Class H ramps 5-10, and (d) the GZW of Class G in inner Stancomb-Wills Trough. For locations of profiles in (a), (b), (c) and (d) see Figures 1, 3, 4 and 6, respectively.**


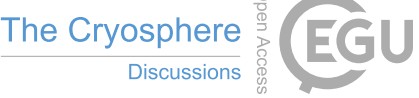

**Figure 6: High-resolution bathymetric map from (a) inner and (b) middle Stancomb-Wills Trough showing ploughmarks eroded by individual icebergs (Class A) and by icebergs pushed by an ice shelf in a preferred orientation (Class B), glacial lineations (long arrows = Class C, long dashed arrows = Class D), and two grounding-zone wedge (GZW) fronts (bold short arrows, Class G). The acoustic subbottom profile (c), whose location is shown in (a), crosses bedforms of Classes A, B, C, D and G.**





**Figure 7: High-resolution bathymetric map (a) from the western slope of McDonald Bank showing curvilinear ridges and grooves of Class I that are interpreted as ice marginal bedforms. Bold short arrow marks the front of the westernmost Class H ramp. Subbottom profile (b) displays the depths of the groove incisions, the lack of penetration of the acoustic signal, and a different acoustic facies northwest of the ridges and grooves.**



**Figure 8: Sedimentological parameters and radiocarbon ages from cores in the study area (for location, see Fig. 2): a) Lithology,**
**shear strength (str.), wet bulk density (WBD), magnetic susceptibility (mag. sus.), grain-size composition of sediment matrix, water**
**content (H₂O), facies interpretation and AMS ¹⁴C ages of core GC635 (this study); b) Lithology and AMS ¹⁴C ages of core 3-7-1**
**(modified from Stolldorf et al., 2012). For ages, see Table S2.**





**Figure 9:** a) Landsat satellite image (courtesy of the U.S. Geological Survey) showing the location of four large ice slabs enclosed
within perennial sea ice in February 2017 and their trajectory since 1973. b) Map of sub-ice shelf bathymetry, inferred from gravity
inversion in the central part (Hodgson et al., 2019), with locations of mega-scale glacial lineations (Class C and D) and iceberg ramps
(Class H). Furthermore, the main trajectory of large ice slabs under modern conditions (see Fig. 9a) and the assumed main trajectory
of paleo-ice slabs at the time of ramp formation are shown. The inferred direction of paleo-ice streaming support the continuation
of Stancomb-Wills Trough into a modern subglacial trough. c) Profiles of ice surface, bedrock topography (Fretwell et al., 2013) and
ice-flow speed (Mouginot et al., 2017) inshore of the grounding line (for location, see Fig. 9a). Note the high flow velocity in the



**deepest section flowing into Stancomb-Wills Ice Tongue and the shallow bedrock altitude with thick ice in the suture zone forming the large ice slabs.**

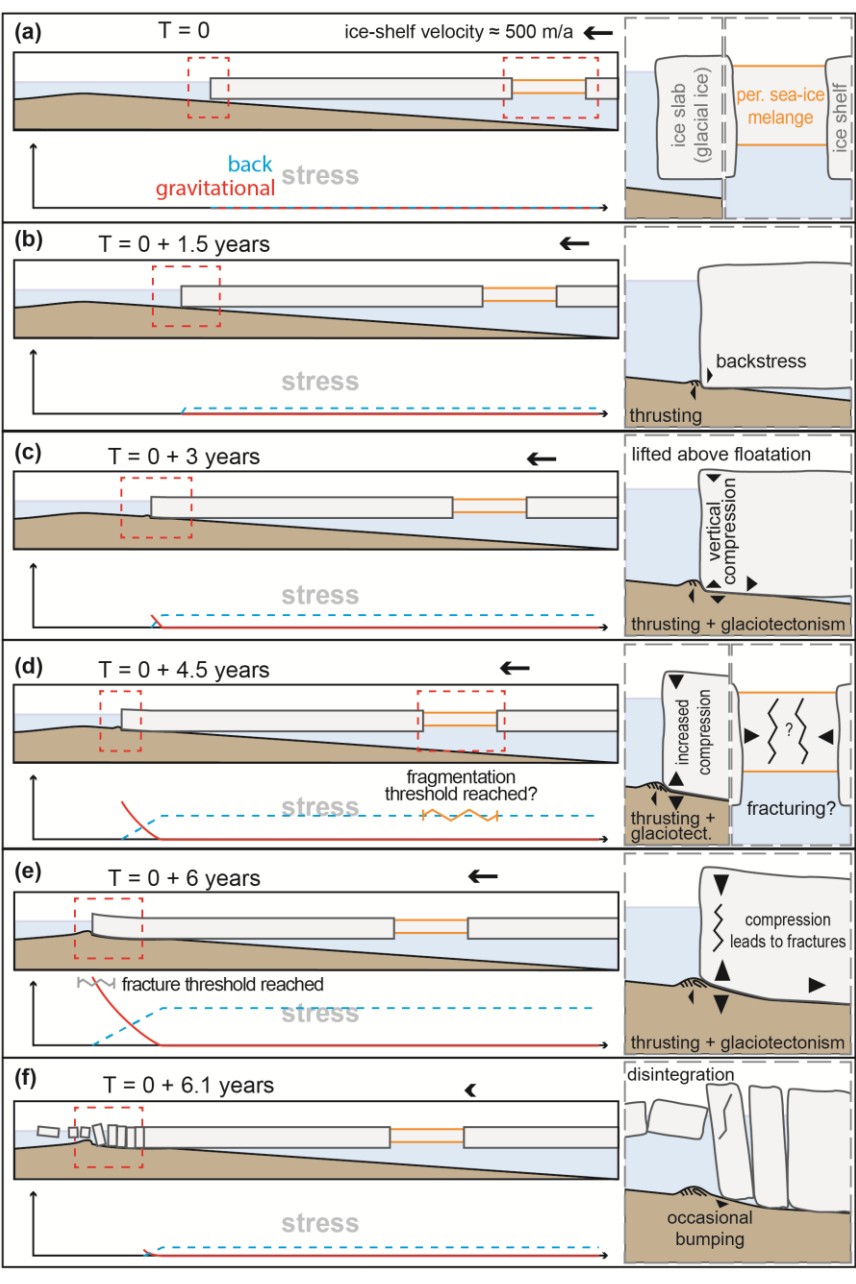

**Figure 10: Sketch of the proposed iceberg ramp (Class H) formation process, including a conceptual graph of stresses active during formation, with time estimates based on modern ice shelf flow velocity of about 500 m/a. a) Setting before the contact of the ice slab with the seafloor. b) Initial contact of the ice slab with the seafloor leads to seaward directed thrusting of sediments at the front of the slab and build-up of back stress within the ice slab further upstream of the point of contact. c) The front of the ice slab is 'jacked up' causing gravitational stresses and vertical compression within the ice slab; continued thrusting of the sediments combined with their compression leads to glaciotectonism; back stress within the slab increases further. d) Continued uplift of the slab front increases stresses, with the back stress probably exceeding a threshold that causes the fragmentation of the perennial sea-ice**



melange. e) Gravitational stress reaches a level that causes fracturing of the ice slab front. f) The ice slab disintegrates into smaller icebergs, which capsize and occasionally 'bump' into the top of the newly formed ramps when they drift seawards.

**Table 1: Expeditions that collected investigated swath bathymetry data within the research area and the used acquisition systems.**

| Expedition | Vessel | Acquisition System |
|---|---|---|
| ANT-IV/3 (1985/86), ANT-V/4(1986/87), ANT-VI/3 (1987/88) | RV *Polarstern* | Seabeam, 12.3 kHz, 16 beams |
| ANT-VIII/5 (1989/90), ANT-IX/3 (1991), ANT-X/2 (1992), ANT-XII/3 (1995) | RV *Polarstern* | Hydrosweep DS1, 15.5 kHz, 59 beams |
| ANT-XV/3 (1998), ANT-XVI/2 (1999) | RV *Polarstern* | Hydrosweep DS2, 15.5 kHz, 59 beams |
| PS82 (2013/14), PS96 (2015/16), PS111 (2018) | RV *Polarstern* | Hydrosweep DS3, 15.5 kHz, 345 beams |
| JR97 (2005), JR206 (2010), JR244 (2011), JR259 (2012) | RRS *James Clark Ross* | EM120, 11.25-12.75 kHz, 191 beams |






**Table 2: Morphological classes identified in the swath bathymetric data and their properties (SWT: Stancomb-Wills Trough).**

| Class | Shape | Orientation (west of north) | Amplitude/ Height (m) | Width (m) | Depth Range (m) | Interpretation | Figure |
|---|---|---|---|---|---|---|---|
| A | single, linear to curvilinear furrows (locally with berms on the sides) | randomly oriented | 3-12 | 100 – 650 | < 500 (locally < 600) | Ploughmarks of single icebergs | Fig. 6b |
| B | furrows with preferred orientation | around 55° - 65° | 2-8 | 100 – 300 | 500 – 600 | Ploughmarks of icebergs pushed by ice shelf | Fig. 6 |
| C | parallel, linear ridges | 55° - 65° | 1-6 | 150-500 | 550 – 700 | Mega scale glacial lineations | Fig. 6 |
| D | parallel, linear ridges | ~40° | 1-3 | ~ 200 | 600 – 650 | Mega scale glacial lineations | Fig. 6a |
| E | mostly parallel, locally curvilinear sets of ridges | around 50° - 105° | 3-10 | 150 – 300 | 250 – 500 | Squeezed and smeared ridges of grounded ice slabs enclosed in perennial sea-ice | Fig. 4 |
| F | wedge shaped bank | NW-SE | 110 | ~ 15000 | 280 – 440 | lateral grounding-zone wedge | Fig. 5a |
| G | wedge shaped sills, with steep (> 3°) sinuous front | fronts meandering from north to south | 10 – 20 | > 2000 | 500 – 630 | grounding-zone wedge | Fig. 6a |
| H | wedge shaped ramps, some horseshoe-shaped in plan view | approx. pointing to the west | 20 – 50 | > 2000 | 200 - 500 | Formed by grounding meteoric ice slabs enclosed in perennial sea-ice | Fig. 3 and 4 |
| I | curvilinear ridges and grooves | NNE to SSW | 5 - 20 m | 500 - 1000 m | 280 - 410 m | Ice-marginal landform | Fig. 7 |