# Peer review of "Past ice sheet-seabed interactions in the northeastern Weddell Sea Embayment, Antarctica"

_The Cryosphere, 2019_

## Referee Comment (RC1) · Frank Nitsche (Referee) · 16 Jan 2020

General comments:

The manuscript presents a new compilation and interpretation of new and older multi-beam bathymetry data from the continental shelf near the Brunt Ice Shelf in the Weddell Sea. It provides the first detailed past ice sheet reconstruction for this area and discusses processes forming some specific, unusual morphological features. The detailed paleo-ice sheet history for large parts of the Weddell Sea is still uncertain and therefore, this is an important contribution to this discussion. In addition, the observations and discussion of features formed outside fast ice streams. The paper is well written and structured, and I have only some minor comments:

[Figure]

Specific comments:

The observed ramps (type H) and related ridges (type E) are discussed in detail and are a centerpiece of the paper. It would be good to also compare these features to the ones described by Jakobssen et al. 2011 (https://doi.org/10.1130/G32153.1). They describe the movement of melange of broken-up ice shelf in Pine Island Bay including smaller iceberg plow ridges.

Line 268: Could the ramps 5 and 7 actually be some kind of GZW? It seems possible that the GZW in the top right of figure 4 extended along H5 and H7. The absence of MSGL in this part likely indicates lower ice movement, as stated in the text, so these GZW would receive less sediment and be smaller. It seems that the events forming the ramp6 is overprinting any older features.

Technical comments:

Line 111/112: The text states that x-radiographs were taken every 2 cm. Aren't X-radiographs usually combined into a continues image? Are the 2cm a resolution or a stepping interval?

Line132: It might be worth mentioning that the ice slap is not moving fast, so 14/17 year intervals are fine.

Line 226: add space between "landward" and "shelf"

Line 288: add space between "such" and "topographic"

Line 291: Maybe use a different phrase instead of "supposed" here. Maybe something like "These studies suggest that hill-hole pairs occur . . ."

Line 413: add space between "stresses" and "that"

Line 471: Is the reference to figure 8a is correct here? It seems Fig 9b would be more appropriate here.

[Figure]

Line 530: Specify the "most ice of this drainage area ..." to avoid misunderstanding that most ice of the EAIS was flowing through the study area.

Line 779, Figure 9: Add a note that the black line represents modern ice and ice shelf edge. Line 780: change "see Fig. 9a" to "see blue line in Fig 9b"

---

## Referee Comment (RC2) · Chris Clark (Referee) · 30 Jan 2020

Overview The paper is exceptionally well written and presented and shows some well-known landform types on the seafloor that are used to build a reconstruction of the ice stream and ice shelf and their flow pattern and retreat. Some dating control is presented to anchor the reconstruction in time. This part of the paper forms a useful and important contribution to the growing field of glacial history surrounding Antarctica. The paper also shows some landform systems that appear to differ and are therefore thought of as a new type. Using the modern context of an ice shelf with icebergs glued together by sea ice, they build a conceptual interpretation suggesting how these might have been formed. This aspect is more speculative, but interesting and is worth presenting, although I nudge at it with an alternative view in case the authors think it is

worth noting. I make a suggestion for a further figure, which would help the impact of the paper.

Abstract and introduction do a good and comprehensive job of setting up the context from the literature and the geographic and glaciological backgrounds.

Section 4.2. The interpretation here is supported by the observations and presumably sensibly influenced by the modern observations of icebergs in a melange of sea ice. This is neat. In this story I suppose icebergs calve off and rotate and once buoyant produce angular keels that can accomplish ploughing some grooves. Upstream of these parallel grooves however (in fig 6) are ridge-grooves that you call MSGL. It is striking that they look similar in orientation and scale, with MSGL possibly leading into iceberg grooves (maybe you could check in your data). I wonder if worth considering if the keels transited the whole system and that is not necessary for individual icebergs to make the grooves; ie keels made the MSGL and the class B ploughmarks. This is something I noted in the Norwegian Channel ice stream making the link that the keels produced both landforms across a transition from grounded ice to lightly grounded to floating ice(Clark et al' 2003 J- Glacial see fig 11 which looks similar to me). This is not critical to your reconstruction but I suggest worth asking if the grooves can actually be traced from one type to the other. This interpretation might also be relevant for your class H ramps – and class E grooves (fig 4) in section 4.4 rather than basal crevasses and the shearing. Perhaps all the lineations (C, D, B, and E) are the same thing but with different degrees of grounding. Not all of you E landforms are sinuous and in fact look very similar to MSGL in places. In this interpretation, you class H ramps and its smeared ridges are the same thing as GZW with grooved MSGL leading to iceberg grooves. The only difference being orientation of ice flow and perhaps ice velocity. I mention all this as an alternate interpretation in case you want to consider it.

Section 5.2.3. The reconstruction of ice stream, ice shelf, glued icebergs and their extent and geometry is described and I could visualise with reference to the map in Fig 2. I would consider making this easier for the reader by providing a new figure

showing the reconstructed cartoon-map of palaeo geography that you describe. This could make the paper more accessible to those interested in the glacial history and increase its impact.

Minor points In abstract it is stated that S- W Trough was main drainage for EAIS. Clarify if this is what you mean ie. whole ice sheet or if you mean in your studied sector.

Line 155 and many places elsewhere, probably better to call them landforms rather than bedforms. To many, the latter term has a more restricted use such as MSGL, drumlins etc.

Para 415 stresses that - space needed 470 Fig 96 not 8 a

Chris Clark (30 Jan)

---

## Author Response (AR1)

**Point by point response to the reviews**

**In the following we mark reviewer comments (*in italic*) with (1), our responses (bold) with (2), and our changes (bold) to a revised manuscript with (3):**

*Comments of reviewer 1 (Frank Nitsche):*

*(1) General comments:*
*The manuscript presents a new compilation and interpretation of new and older multibeam bathymetry data from the continental shelf near the Brunt Ice Shelf in the Weddell Sea. It provides the first detailed past ice sheet reconstruction for this area and discusses processes forming some specific, unusual morphological features. The detailed paleo-ice sheet history for large parts of the Weddell Sea is still uncertain and therefore, this is an important contribution to this discussion. In addition, the observations and discussion of features formed outside fast ice streams. The paper is well written and structured, and I have only some minor comments:*

**(2) We thank Frank Nitsche for this positive review.**

**(1)** *Specific comments:*
*The observed ramps (type H) and related ridges (type E) are discussed in detail and are a centerpiece of the paper. It would be good to also compare these features to the ones described by Jakobssen et al. 2011 (https://doi.org/10.1130/G32153.1). They describe the movement of melange of broken-up ice shelf in Pine Island Bay including smaller iceberg plow ridges.*

**(2) We agree that a comparison to this proposed process should be added. We referred to the iceberg plow ridges observed by Jakobsson et al. (2011) in the section on Class H formation as terminal berms (line 325), but so far did not mention that these were hypothesized to have formed by the mélange of a broken up ice-shelf. This process indeed has some similarities to the process we propose for offshore the Brunt Ice Shelf, but occurred at different scale and in a different geographical setting.**
**In addition, we used the iceberg ploughmarks size values of a statistical investigation of swath bathymetry from the eastern Amundsen Sea (Wise et al. 2017) for testing, whether Class H ramps could represent such iceberg berms (line 327). This investigation analysed the data presented by Jakobsson et al. (2011).**

**(3) We added information on the proposed formation process in the Amundsen Sea, including its differences to our Class H landforms, to the revised version.**

**(1)** *Line 268: Could the ramps 5 and 7 actually be some kind of GZW? It seems possible that the GZW in the top right of figure 4 extended along H5 and H7. The absence of MSGL in this part likely indicates lower ice movement, as stated in the text, so these GZW would receive less*

*sediment and be smaller. It seems that the events forming the ramp6 is overprinting any older features.*

**(2) Ramps 5 and 7 are the most ambiguous landforms of Class H ramps with the highest similarities to GZWs. Our main reasoning for classifying them as Class H ramps and not as GZWs was the close proximity to other Class H features (line 272), the lack of MSGLs (line 277) and their setting outside of a palaeo-ice stream trough (line 278). Nevertheless, we cannot rule out the possibility that ramps 5 and 7 represent GZWs in a situation as described by the reviewer.**

**(3) We clarify in the revised version of the manuscript that we do not rule out a GZW origin for ramps 5 and 7.**

**(1)** *Technical comments:*
*Line 111/112: The text states that x-radiographs were taken every 2 cm. Aren't Xradiographs usually combined into a continues image? Are the 2cm a resolution or a*
*stepping interval?*
*Line132: It might be worth mentioning that the ice slap is not moving fast, so 14/17*
*year intervals are fine.*
*Line 226: add space between "landward" and "shelf"*
*Line 288: add space between "such" and "topographic"*
*Line 291: Maybe use a different phrase instead of "supposed" here. Maybe something like "These studies suggest that hill-hole pairs occur : : :"*
*Line 413: add space between "stresses" and "that"*
*Line 471: Is the reference to figure 8a is correct here? It seems Fig 9b would be more appropriate here.*

**(2) We thank the reviewer for these valid technical comments.**

**(3) We changed the manuscript accordingly.**

*Comments of reviewer 2 (Chris Clark):*

**(1)** *Overview The paper is exceptionally well written and presented and shows some wellknown landform types on the seafloor that are used to build a reconstruction of the ice stream and ice shelf and their flow pattern and retreat. Some dating control is presented to anchor the reconstruction in time. This part of the paper forms a useful and important contribution to the growing field of glacial history surrounding Antarctica. The paper also shows some landform systems that appear to differ and are therefore thought of as a new type. Using the modern context of an ice shelf with icebergs glued together by sea ice, they build a conceptual interpretation suggesting how these might have been formed. This aspect is more speculative, but interesting and is worth presenting, although I nudge at it with an alternative view in case the authors think it is worth noting. I make a suggestion for a further figure, which would help the impact of the paper.*

**(2) We are grateful to Chris Clark for his evaluation of our study and his suggestion for a further figure.**

*Abstract and introduction do a good and comprehensive job of setting up the context from the literature and the geographic and glaciological backgrounds.*

**(1)** *Section 4.2. The interpretation here is supported by the observations and presumably sensibly influenced by the modern observations of icebergs in a melange of sea ice. This is neat. In this story I suppose icebergs calve off and rotate and once buoyant produce angular keels that can accomplish ploughing some grooves. Upstream of these parallel grooves however (in fig 6) are ridge-grooves that you call MSGL. It is striking that they look similar in orientation and scale, with MSGL possibly leading into iceberg grooves (maybe you could check in your data). I wonder if worth considering if the keels transited the whole system and that is not necessary for individual icebergs to make the grooves; ie keels made the MSGL and the class B ploughmarks. This is something I noted in the Norwegian Channel ice stream making the link that the keels produced both landforms across a transition from grounded ice to lightly grounded to floating ice(Clark et al' 2003 J- Glacial see fig 11 which looks similar to me). This is not critical to your reconstruction but I suggest worth asking if the grooves can actually be traced from one type to the other. This interpretation might also be relevant for your class H ramps – and class E grooves (fig 4) in section 4.4 rather than basal crevasses and the shearing. Perhaps all the lineations (C, D, B, and E) are the same thing but with different degrees of grounding. Not all of you E landforms are sinuous and in fact look very similar to MSGL in places. In this interpretation, you class H ramps and its smeared ridges are the same thing as GZW with grooved MSGL leading to iceberg grooves. The only difference being orientation of ice flow and perhaps ice velocity. I mention all this as an alternate interpretation in case you want to consider it.*

**(2)** These are interesting thoughts and the mapped parts of transition from Class C to Class B features share some similarities with the features in Figure 11 of Clark et al. (2003). However, the main problem preventing us from investigating this further is the lack of data in the zone, where these features transition into each other. Figures 6a and 6b present all the swath bathymetric data available from regions, where such transitions might occur.

In Figure 6a only few Class B features are present. These either do not show a continuation upstream, or swath bathymetric data from further upstream is unavailable due to ice-shelf coverage. In Figure 6b, where Class C features are identified, no Class B features are observed directly downstream in inferred ice flow direction. This area looks rather "flattened" or smeared, probably due to sediment reworking by other icebergs at a later stage. Class B features occur in Figure 6b, i.e further downstream and closer to the northern end of the Stancomb-Wills Trough, but again no swath bathymetric data is available directly upstream due to permanent/regular ice shelf coverage of this area. Therefore it is not possible to check, if these Class B features continue upstream, become more linear/parallel and turn into Class C features.

The same problem of swath bathymetric data lacking upstream due to ice-shelf coverage exists for most of the Class E grooves (see Figure 4). For the few Class E grooves, where data is available upstream, no continuation into MSGLs is observed. Therefore, also in this case it is either impossible to check the hypothesis, that Class E grooves develop from MSGLs, or such a transition is not observed.

In summary, where data is available upstream of Class B and Class E features, we were not able to trace these back to MSGLs, but in most cases ice-shelf coverage is responsible for the lack of data, making it impossible to trace these features further upstream. Therefore, our data unfortunately is not suitable for verifying the hypothesis of these features evolving into each other.

**(3) We decided not to incorporate a related discussion in the manuscript because a) our data either do not show such a transition between different feature classes or are unsuitable for investigating this, and b) this is not critical for our further interpretations, as it is also stated by the reviewer.**

*(1) Section 5.2.3. The reconstruction of ice stream, ice shelf, glued icebergs and their extent and geometry is described and I could visualise with reference to the map in Fig 2. I would consider making this easier for the reader by providing a new figure showing the reconstructed cartoon-map of palaeo geography that you describe. This could make the paper more accessible to those interested in the glacial history and increase its impact.*

**(2) We agree that a visualisation of this reconstruction would be very valuable for communicating our thoughts to the reader. Initially, we refrained from creating such a cartoon because the chronology of the two described glaciological regimes is not clearly**

resolved (see lines 512-522). Therefore, we thought such a figure may oversimplify our knowledge of the glacial history in the study area and thus may potentially mislead the reader.

**(3) We created a schematic cartoon illustrating the two reconstructed glaciological regimes in the study area and added this as a new figure to the revised manuscript.**

*(1) Minor points*
*In abstract it is stated that S-WTrough was main drainage for EAIS. Clarify if this is what you mean ie. whole ice sheet or if you mean in your studied sector.*
*Line 155 and many places elsewhere, probably better to call them landforms rather than bedforms. To many, the latter term has a more restricted use such as MSGL, drumlins etc.*
*Para 415 stresses that - space needed*
*470 Fig 96 not 8 a*

**(2) We agree with the reviewer on all his 'minor points'.**

**(3) We changed the manuscript accordingly.**

**List of all relevant changes made in the manuscript**

1. We added information on the proposed formation process of iceberg plow ridges in the Amundsen Sea, as suggested by reviewer Frank Nitsche.
2. We clarified that we do not rule out a GZW origin for ramps 5 and 7, as suggested by reviewer Frank Nitsche.
3. We created a schematic cartoon illustrating the two reconstructed glaciological regimes in the study area, as suggested by reviewer Chris Clark, and added this as a new figure (Figure 11).
4. In addition, we made some minor changes to the manuscript, including those remarked by the reviewers, and updated the references.

[revised manuscript text omitted]

**Trough in the eastern Amundsen Sea were created by an iceberg mélange resulting from the break-up of an ice shelf. These landforms and their proposed formation process are, however, not a blue print for the Class H ramps in front of the Brunt Ice Shelf due to significant differences. The Class H iceberg ramps are approximately five times larger than the largest Pine Island Trough landforms. Furthermore, they occur outside of a prominent paleo-ice stream trough (characterized by abundant presence of MSGLs), and their occurrence on a bed interpreted as having been covered by**

**cold-based ice points towards low availability of sediment (i.e. lack of soft till). Nevertheless, the suggested formation process for the Pine Island Trough landforms indicates that larger-scale landforms may be created by icebergs, if external forces, in this case the pressure of an iceberg mélange, are active.**

[revised manuscript text omitted]